

# Quantitative soil characterization using frequency domain electromagnetic induction method in heterogeneous fields

Gaston Matias Mendoza Veirana[1], Guillaume Blanchy[2,3], Ellen Van De Vijver[1], Jeroen Verhegge[1,4],
Wim Cornelis[1], Philippe De Smedt[1,4]

[1]Department of Environment, Faculty of Bioscience Engineering, Ghent University, Coupure Links 653, geb. B, 9000 Ghent, Belgium.

[2] Urban and Environmental Engineering, University of Liège, Liège, Belgium,

[3] F.R.S-FNRS (Fonds de la Recherche Scientifique), Brussels, Belgium

[4] Department of Archaeology, Ghent University, Sint-Pietersnieuwstraat 35-UFO, 9000 Ghent, Belgium.

*Correspondence to*: Gaston Matias Mendoza Veirana (Gaston.MendozaVeirana@ugent.be)

**Abstract.** The frequency domain electromagnetic induction (FDEM) method is a widely used tool for geophysical soil exploration. Field surveys using FDEM provide apparent electrical conductivity (ECa), which is typically used for qualitative interpretations. Quantitative estimations of soil properties remain challenging, especially in heterogeneous fields. Quantitative approaches are either based on deterministic or stochastic modeling. While the deterministic approach faces limitations related to instrumental drift, data calibration, inversion, and pedophysical modeling, the stochastic approach requires developing a local model, which involves extensive field sampling.

This study aims to evaluate the effectiveness of the FDEM modelling based on either a deterministic or stochastic approach, identify its limitations, and search for optimal field protocols. We provide practical guidelines for end-users to quantitatively predict soil water content, bulk density, clay content, cation exchange capacity, and water EC in heterogeneous fields.

Two field surveys were conducted in Belgium, where FDEM data was collected using Dualem-421S and Dualem-21HS sensors, along with data taken from electrical resistivity tomography (ERT) measurements and an impedance moisture probe, and soil sampling.

A comprehensive sensitivity analysis revealed that deterministic modeling procedures could not predict water content more accurately than a mean value approximation (negative R[2]). This analysis also highlighted the sensitivity of the minimization method used in FDEM data inversion and the applied pedophysical model. Stochastic modeling, which does not require FDEM data calibration or inversion, outperformed the deterministic approach. However, its prediction accuracy is limited, particularly if soil sample depth is not considered.

## 1 Introduction

Frequency domain electromagnetic induction (FDEM) tools are widely applied in geophysical soil surveys (Boaga, 2017). These instruments often serve to qualitatively determine spatiotemporal changes in the apparent electrical conductivity ($EC_a$), reflecting the influence of soil characteristics within the measured soil volume (Doolittle and



Brevik, 2014). As the relationship between electrical conductivity ($EC$) and several of such soil attributes has been investigated extensively, FDEM is also capable of their quantitative assessment. Specifically, the soil water content is a preferred target because of its central role in soil-plant interaction, groundwater assessment, soil ecological functioning, and climate regulation.

        Despite these applications, a broader practical implementation of FDEM remains mainly limited to academic settings
(Altdorff et al., 2017; Huang et al., 2007). Two major challenges hinder wider adoption. Firstly, the FDEM methodology itself faces issues such as instrumental drift, approximations to translate raw FDEM data to $EC_a$, calibration difficulties (Hanssens et al., 2020; Minsley et al., 2012), and the necessity for data inversion of $EC_a$ to true $EC$ before quantitative assessments can be made. This reality persists even though – particularly for research purposes – adaptive correction procedures and open-source inversion codes have become available. Secondly, a significant
obstacle in translating soil $EC$ data into a target soil property lies in the current limitations of pedophysical models. These models are deterministic and link geophysical variables with soil properties (see e.g., Glover, 2015; Romero-Ruiz et al., 2018), but often suffer from a lack of precision and are difficult to generalize. This is exacerbated by the variability in soil types, spatial heterogeneity, temperature conditions, the electromagnetic frequency of measurements (Moghadas and Badorreck, 2019), and the difference between the laboratory-analyzed and FDEM-measured soil
volumes. As an alternative to pedophysical models, field-specific stochastic relationships can be composed at the cost of obtaining significant amounts of calibration data (Corwin and Lesch, 2003). Despite stochastic modelling is inherently limited to the conditions represented by the dataset the model has been trained for, exhaustive assessments of this method demonstrated useful predictions of various soil properties across agricultural fields (Boaga, 2017; Rentschler et al., 2020).

Here, we evaluate how FDEM data can serve to quantitatively predict spatial variations in volumetric soil water content ($\theta$), bulk density ($\rho_b$), clay content, cation exchange capacity ($CEC$), and water $EC$ ($EC_w$) in a practical, straightforward manner on two heterogeneous test sites. In search for optimal field protocols, we evaluate to which extent considering instrumental limitations and different procedures of FDEM data correction and processing influence the accuracy of the predicted soil attributes, and what the trade-off between deploying a physics-driven
deterministic versus field-specific data-driven model implies. Finally, we propose field and modelling strategies for optimal soil characterization with FDEM surveys.

        **2 Methodology**

**2.1 FDEM functioning**

        FDEM devices function by transmitting an alternating current through a transmitter coil, creating a primary electromagnetic field that varies over time. This primary field interacts with the subsurface, inducing eddy currents which subsequently produce a secondary electromagnetic field. Both fields are detected by the receiver coil and are
quantified as a complex number, consisting of an in-phase component (IP) and a quadrature component (QP). The ratio of these fields, reflecting both the device setup and the conditions of the subsurface, is typically measured.



The field ratio (in ppm) can be converted to the actual $EC_a$ by using the linear model developed by McNeill (1980) assuming a homogeneous subsurface electrical conductivity. This model assumes a uniform subsurface $EC$ and is known as the low induction number (LIN) approximation. It is valid when the induction number (β) is low (β ≪ 1). The LIN approximation proposed by McNeill (1980) is given by:

$$EC_a = QP \frac{4}{\mu_0 \omega s^2} \text{ when } \beta = s\sqrt{\mu_0 \omega EC}/_2 \ll 1$$

*Equation 1*

where $\omega$ is the angular frequency, $\mu_0$ is the magnetic permeability of free space ($1.257 \ 10^{-8}$ H/m) and $s$ is the coil separation. It can be seen from this equation that large frequencies and higher $EC$ soils will violate the β ≪ 1 specification. It is important to note that the LIN approximation also assumes that the FDEM device is operated at ground level above a homogeneous, poorly conductive subsurface (Callegary et al., 2007; McNeill, 1980).

**2.2 Data collection**

Two heterogeneous agricultural fields were examined in this study. Site 1, located in Middelkerke, Belgium, is shown in Figure 1A. Belgian soil map data (Van Ranst & Sys, 2000) indicate that the field is affected by saline groundwater and exhibits a soil texture varying from loam (26% clay, 34% sand) to silt loam (10% clay, 40% sand) (USDA textures), with clay layers starting at depths greater than 0.50 m. In contrast, Site 2, located in Bottelare, Belgium (Figure 1B), is characterized by fresh groundwater. The soil texture at this location ranges from sandy loam (13% clay, 76% sand) to clay (64% clay, 5% sand).

Field surveys at both sites involved collecting FDEM data using different sensors, all operating at 9 kHz: the Dualem-421S at Site 1 with a 3 m crossline sampling density, and the Dualem-21HS at Site 2 with a 1 m crossline sampling density, both with a distance above ground of 0.165 m. Driving speed was approximately 10 km/h with a measurement sampling rate of 10 Hz. The crossline density was decided based on the time to survey each field, being Site 1 bigger.

The surveys at both sites provided in-phase (IP) and quadrature phase (QP) data with an in-line sampling density of approximately 0.3 m in horizontal co-planar (HCP) and perpendicular (PRP) configuration. For both sites, transmitter-receiver separations of 1.0 m (HCP1.0), 1.1 m (PRP1.1), 2.0 m (HCP2.0), and 2.1 m (PRP2.1) were used. Additionally, 4.0 m HCP (HCP4.0) and 4.1 m PRP data (PRP4.1) were collected at Site 1, and 0.5 m HCP (HCP0.5) and 0.6 m PRP data (PRP0.6) at Site 2. Electrical resistivity tomography (ERT, Syscal Pro, Iris Instruments) was performed with 0.5 m electrode spacing, during the FDEM surveys (Figure 1). The ERT transects were located based on previous surveys to include the largest $EC_a$ range across the field.

In addition to geophysical surveys, soil sampling was carried out at 15 predetermined locations at each site. These locations were strategically selected using the Latin Hypercube Sampling method (Minasny & McBratney, 2006), and were based on insights from previously collected FDEM data. Undisturbed soil samples were extracted from two depths, 0.10 m (topsoil) and 0.50 m (subsoil) below the surface, in stainless steel 100-cm³ cores using an auger. In total 30 samples per site were analyzed in the laboratory to obtain soil texture (after sieving at 2 mm), $CEC$ (CoHex method (Ciesielski et al., 1997a, 1997b)), $\theta$ and $\rho_b$ (gravimetric method with convective oven drying at 105 °C).



To accurately determine in-situ $EC$, $EC_w$, and temperature within the soil sampling volume (100-cm$^3$), measurements were taken at each sampling location using a HydraProbe soil probe (HydraProbe, Stevens Water Monitoring Systems, 2008). The correction proposed by Logsdon et al. (2010) was applied to improve the quality of these $EC$ readings.

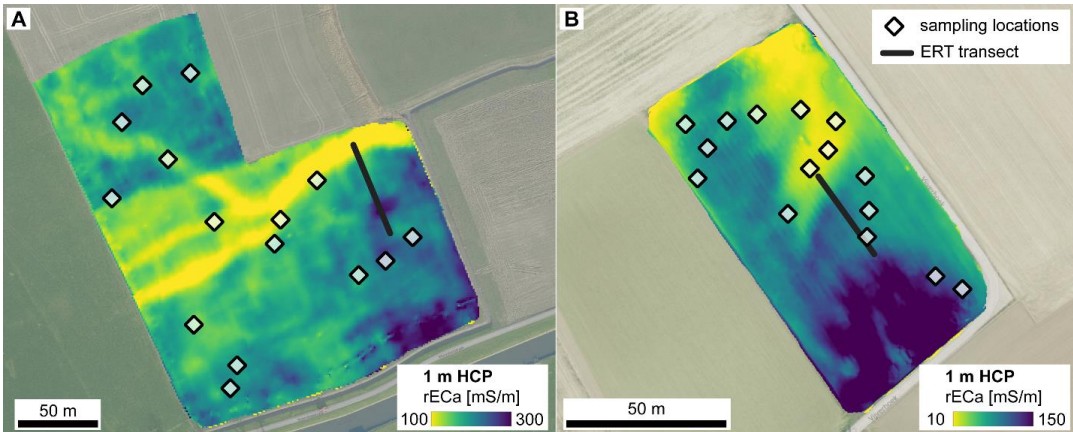

**Figure 1 Site 1 in Middelkerke [A] and Site 2 in Bottelare [B] (Belgium), with the position of the ERT transects and 15 soil sampling locations per site, selected via conditioned Latin Hypercube Sampling based on previously obtained FDEM LIN $EC_a$ data. The mapped data is ERT-calibrated robust $EC_a$ HCP1.0.**

### 2.3 Data processing

The general processing workflow of the FDEM survey follows Hanssens et al. (2020), and is described in Figure 2. The methodology aims at processing the FDEM data to obtain reliable $EC$ data at sampling locations and then predict soil properties. For this process, a meaningful physical modeling sequence was followed. For instance, no inversion was implemented on uncalibrated FDEM data. This involved four key steps: ERT inversion, FDEM data calibration, FDEM data inversion, and pedophysical modeling (Hanssens et al., 2020). All computer code used is open-source, and default parameters were prioritized, ensuring reproducibility of methodology and results. All developed codes for this section are available at Mendoza Veirana (2024b), and collected data at Mendoza Veirana (2024a).








**Figure 2 Workflow of geophysical data process including prediction of soil properties. Ellipses represent observations that exist independently of all data processes. Rectangles represent data over all the field and/or soil sampling locations. Parallelograms represent data over the ERT profiles. The square represents an external model. Colors represent modelling processes: light blue for ERT inversion (Jupyter Notebook '00_inv-ERT'), green for FDEM data calibration (Jupyter Notebook '01_QP_cal'), orange for FDEM data inversion (Jupyter Notebook '02_EC_invert'), and red for soil properties modelling (Jupyter Notebook '03_Soil_properties_modelling').**

### 2.3.1 ERT inversion

The measured ERT data was inverted using the ResIPy (v3.5.4) open-software (Blanchy et al., 2020) which is based on the R2 codes (Binley & Kemna, 2005) (see full code in the Jupyter Notebook '00_inv-ERT'). A standard inversion using a triangular mesh was implemented, converging after three iterations. After inversion, extraction of ERT profiles was done by averaging the $EC$ in a neighborhood of 0.5 m around each electrode. Alternatively, to obtain smoother profiles an extraction window of 2.5 m was also used.

### 2.3.2 FDEM data calibration

Calibrating raw FDEM data is required for obtaining reliable $EC$ data at sampling locations, and such calibration was done by combining ERT and FDEM data (Lavoué et al., 2010; van der Kruk et al., 2018). On the one hand, the raw uncalibrated FDEM data in parts per thousand (ppt) was transformed to $EC_a$ data following the low induction number (LIN) approximation. On the other hand, the inverted ERT $EC$ data was firstly grouped by profiles and shortened by

removing the profiles at the beginning and end of the transect that do not reach a minimum depth of 4 m due to lower sensitivity on the edge of the transect. After this, 100 profiles remained for Site 1 and 40 profiles for Site 2. Subsequently, the inverted ERT profiles were forward modeled to the theoretical FDEM LIN $EC_a$ measured by the Dualem instrument over the ERT transect (Lavoué et al., 2010). The forward model implements a 1D full solution of Maxwell's equations considering an electromagnetic field, which after FDEM instrument reading, is composed by IP

and QP signals.

Once both FDEM LIN $EC_a$ data were calculated over the ERT profiles, they were matched by spatial proximity with the closes FDEM datapoint, and a linear regression was fitted for the six coil configurations (see Figure 3). Then, this linear calibration was applied to the entire FDEM survey data.



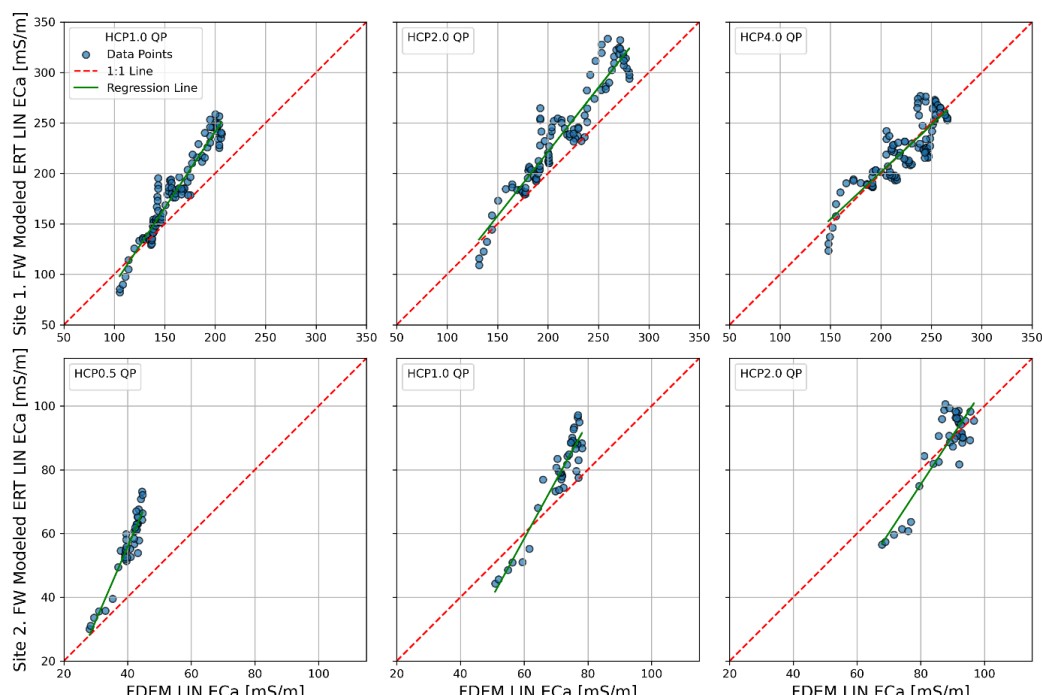


**Figure 3 Calibration of FDEM QP data. A linear regression is fit between the FDEM LIN $EC_a$ data collected on the field (X-axis) over the ERT transect and the FDEM response that was forward modelled from the inverted ERT data (Y-axis). This is shown for the three different QP coil configurations across the three subplot columns, and for both sites displayed in the top and bottom rows.**


Lastly, the calibrated FDEM QP data was transformed to robust $EC_a$ ($rEC_a$) values to provide reliability beyond LIN constraints (Hanssens et al., 2019), such as high salinity and clay content levels. A visual comparison of the forwarded FDEM $EC_a$, uncalibrated, calibrated LIN and $rEC_a$ FDEM data over the ERT transect is shown in Figure 4.





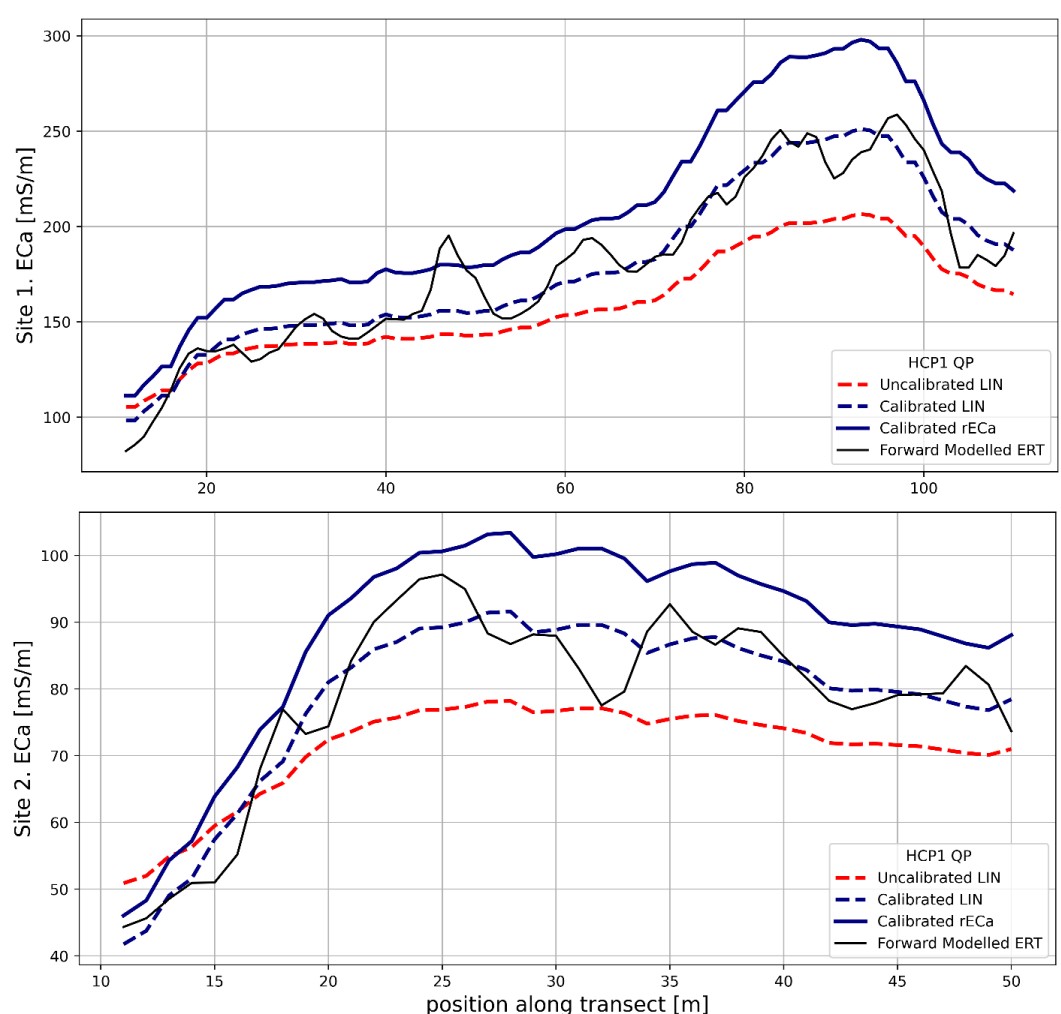

**Figure 4. Comparison of HCP1.0 QP** $EC_a$ **data along the ERT transect for Site 1 (top plot) and Site 2 (subplot). This includes calibrated and uncalibrated FDEM (LIN)** $EC_a$**, and calibrated robust (r**$EC_a$**) data. Also, the forward modelled ERT (LIN)** $EC_a$ **data is shown.**

### 2.3.3 FDEM data inversion

To assess the impact of the different modelling steps, we provide the parameters used and alternatives for comparison. To obtain top- and subsoil $EC$, 1D data inversion was performed with EMagPy (v1.2.2) (McLachlan et al., 2021) using the full Maxwell-based forward model (Wait, 1982). For both sites and based on borehole observations, a five-layer subsurface discretisation was maintained, with fixed interfaces at 0.3 m, 0.6 m, 1.0 m and 2.0 m. Another option for layer interfaces definition consists in using a logarithmic scale from 0.15 m to 2.0 m. The closest FDEM observation to each sampling location was selected as the reference, in contrast to averaging FDEM observations within a radius








(2 m for Site 1, and 1 m for Site 2) around the sampling location. Additional parameters of the inversion problem include an optimization method (Gauss-Newton) (Virtanen et al., 2020) or Robust Parameter Estimation (ROPE) (Bárdossy & Singh, 2008), a vertical smoothing parameter ($\alpha$, default = 0.07), and L2 norm objective function. Moreover, inversion data was composed by all the coil configurations. Removing HCP2.0 and PRP2.1 for Site 1 and

HCP0.5 and PRP0.6 for Site 2 could lead to lower inversion errors. The starting model for inversion was set to the average of the ERT profiles using the given subsurface layers, alternatively, one particular reference ERT profile can be used. Inversions with a negative $R^2$ error were discarded and not analysed further. Finally, $EC$ limits (constraints) were applied to $rEC_a$ FDEM data at sampling locations during its inversion process (just for ROPE solver). These were defined as the minimum and maximum $EC$ values of the inverted ERT profiles.

Once the $EC$ data was obtained by inversion of the $rEC_a$ FDEM data for each sample location, it was used to calculate the soil properties of interest.

### 2.3.4 Soil properties calculation

Linking $EC$ data to soil properties at sampling locations can follow two basic modelling strategies: stochastic and deterministic modelling.

Stochastic modelling enables to empirically predict several soil properties across the surveyed field, at expenses of collecting and analyzing soil samples to build a training dataset. This modelling consists of fitting functions to the training dataset and predicting at targeted locations. Traditionally, polynomial functions have been used for this task

(Rentschler et al., 2020), but in recent years machine learning algorithms (such as artificial neural networks and random forest) have performed better (Moghadas & Badorreck, 2019; Rentschler et al., 2020; Terry et al., 2023). However, using machine learning requires a large amount of training data that may not be obtainable for practical FDEM applications. Thus, we stick to polynomial functions for stochastic modelling.

In our case, for both sites the original soil analysis dataset (n=30) was randomly split into a training dataset (n = 20),

while the remaining was used as test dataset (n=10); this process was repeated 100 times. The optimal polynomial degree was chosen as the one that maximizes the median $R^2$ errors on all the 100 test sets.

Three distinct approaches to polynomial development were utilized. The "Layers Together" for stochastic approach (ST-LT) merged data from different soil depths, treating topsoil and subsoil samples as indistinguishable. In contrast, the "Layers Separate" (ST-LS) approach developed a separate polynomial for each soil layer, using the same number

of samples and applying the same polynomial degree to both topsoil and subsoil data. Finally, the ST-LS2 approach was like ST-LS but permitted different polynomial degrees for the models of each layer.

Model training features included calibrated (LIN and robust) and uncalibrated (LIN) $EC_a$ data from the six FDEM coils, and the inverted $EC$ data at soil sampling locations, while independent targeted soil properties were $\theta$, $CEC$, clay content, $\rho_b$, and $EC_w$.

Deterministic modelling uses general pedophysical $EC - \theta$ relationships that have been validated across a wide range of soil conditions (e.g., models presented by Rhoades et al., 1976). Such modelling does not require calibration data, avoiding the cost of field sampling and laboratory analysis. However, such pedophysical models may fall short in





representing extreme scenarios outside the tested soil characteristics ranges. Additionally, soil data (such as porosity, $EC_w$, and texture) must be available to adequately populate the model and predict the target property. Lastly, soil data

also requires corrections of temperature and electromagnetic frequency (Moghadas and Badorreck, 2019). Because the relationship of $EC$ with soil properties is most straightforward for $\theta$, predicting other targets, such as soil texture or salinity, is generally not feasible under deterministic modelling.

To compare performances between deterministic and stochastic modelling strategies, we tested the pedophysical models on the same test datasets used for stochastic modeling. Three different approaches were employed to populate

the pedophysical model. The deterministic approach for layers together (DT-LT) consisted of averaging soil properties data from all samples regardless of their depth. The layers separate approach (DT-LS) utilized averaged soil properties data from samples at the same layers. The last approach termed the 'ideal' (DT-ID) scenario, used the actual soil properties data from each specific location. Hereby, ideal $EC$ refers to the $EC$ at each sampling location that would result in a perfect $\theta$ prediction after pedophysical modelling.

Predicting $\theta$ via pedophysical modelling followed three steps. First the inverted $EC$ data at 9 kHz were transformed to direct current (DC) $EC$ using the model proposed by Longmire and Smith (1975), which was further validated by Cavka et al. (2014). Then, the resultant DC $EC$ was temperature corrected using the model proposed by (Sheets & Hendrickx, 1995). Lastly, the $EC$ data was converted to $\theta$ based on (Fu et al., 2021):

$$EC = EC_w\theta^2 + \theta\emptyset\left(0.654\frac{clay}{100-clay} + 0.018\right) + (1-\emptyset)EC_s,$$

*Equation 2*

with the solid phase conductivity $EC_s$ (considered negligible), and porosity $\emptyset = 1 - \rho_b/\rho_p$, where $\rho_p$ is the soil particle density ($= 2.65\ \text{g/cm}^3$). All steps were implemented automatically by using Pedophysics open-source software. The pedophysical model of Equation 2 has been validated for samples with 0 to 33% clay content, $\rho_b$ from 1.05 to 1.83 $\text{g/cm}^3$, $EC_w$ from 0.03 to 5.6 S/m, and $\theta$ up to 50 %.

Evaluating the deterministic modelling goodness in comparison with previous studies is not possible because the performance of the FDEM technique is site dependent (Boaga, 2017). Therefore, error indicators ($R^2$ and RMSE) are compared between deterministic and stochastic modelling approaches. Additionally, to assess the limitations of the deterministic modelling, the inverted FDEM $EC$ and ideal $EC$ data are compared to the in-site $EC$ measured with the impedance probe, along with their associated water content.


**2.4 Sensitivity analysis**

In order to develop practical recommendations for FDEM end users and understand the impact of a given parameter (Pannell, 1997), we performed a sensitivity analysis for the most relevant parameters described above. This analysis aimed at finding the impact of alternative choices made during the whole FDEM data processing workflow for

deterministic estimation of water content in the soil samples. The one-at-a-time method, which is the most widely-used sensitivity analysis in environmental sciences (Saltelli & Annoni, 2010) was employed. It consists of altering



one parameter in a stepwise manner and calculating the outcome while fixing other influencing parameters to a predefined origin. Although the one-at-a-time method is practical and easy to implement, it does not give clear information about the effect of all parameters (Saltelli & Annoni, 2010), as the combined effect of two or more parameters is not evaluated. This was solved by deploying the elementary effects method (Saltelli & Annoni, 2010), which consists of changing one parameter at a time, but without returning to an origin. Then by using elementary effects all combinations of parameter´s values were evaluated.

In this study, we defined the origin ($X_0$, see Table 1) as the standard set of parameters used for the whole data process ($F$) that correspond to a standard inversion and subsequent solution ($Y_0$), which is the standard solution for volumetric water content ($\theta_0$):

$$F(Observed\ data, X_0) = Y_0 = \theta_0$$

*Equation 3*

| Parameter | $X_0$ (standard values) | Alternatives |
|---|---|---|
| *Profile extraction distance (m)* | 0.5 | 2.5 |
| *Sample locations* | Closest | Mean |
| *Interfaces* | Observed | Log-defined |
| *Forward model* | FSeq (Full Solution with equivalent EC) | FSlin (Full Solution with LIN approximation), CS (Cumulative Sensitivity) |
| *Minimization method* | Gauss-Newton | ROPE |
| *Smoothing parameter (α)* | 0.07 | 0.02, 0.2 |
| *Remove coils* | False | True |
| *Starting model average* | True | False |
| *Constrain layers EC* | False | True |
| *Deterministic approach* | Ideal | Layers separate, Layers together |

**Table 1. List of model parameters used across all the data workflows. Standard values for each parameter are presented in the second column ($X_0$), and alternatives to these values in the third column.**

## 3 Results and discussion

### 3.1 Comparing EC data

A comparison of $EC$ data obtained by the soil probe observations, standard FDEM inversion (using $X_0$ parameter values), and ideal $EC$ for both sites is shown in Figure 5 alongside the water and clay content of associated samples. The observed water content has mean values of 0.34 and 0.29, and variance of 0.003 and 0.008 for Site 1 and Site 2, respectively.



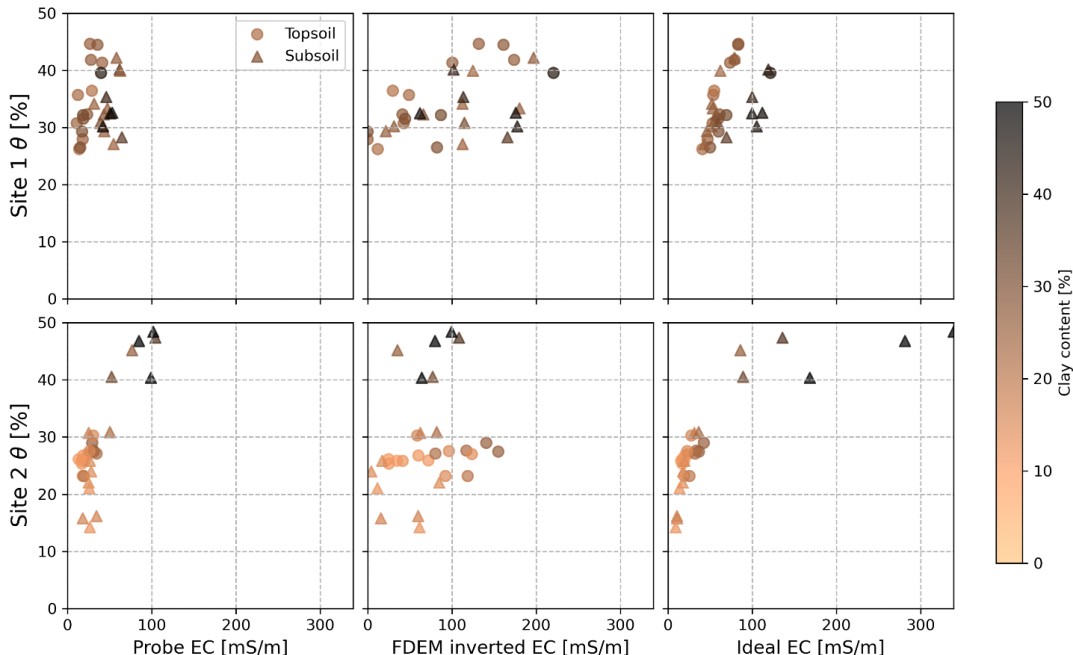


**Figure 5 Comparison between $EC$ obtained with the soil probe (left column), FDEM standard inversion (center), and ideal (right column) versus water content, and clay content as additional dimension. All the $EC$ data is corrected for electromagnetic frequency and temperature (direct current $EC$ at 25 Celsius).**

Considering the $EC$ measured by the soil probe as the reference for actual data, the inverted $EC$ significantly deviates from this reality. Furthermore, while the ideal and soil probe $EC$ display a similar trend, this trend is noticeably stretched (compare first and third column in Figure 5). It is also noteworthy that as the difference between ideal and soil probe EC (for both sites) increases, so does the clay content, with a Pearson correlation of 0.83 (p<0.005), not shown in Figure 5. This disparity becomes even more pronounced for clay contents exceeding ~30%, which is in

accordance with the validity range of Equation 2 (clay contents up to 33%).

### 3.3 Stochastic modelling results

The performance of stochastic models for predicting observed soil properties is presented in Figure 6. Poor predictions

(negative median $R^2$ over test datasets) were obtained when considering topsoil and subsoil data jointly (LT) for model development over training datasets (i.e., not considering sample depth). This may be due to an oversimplistic modelling that does not consider samples depths. Approaches LS and LS2, which use different fitted functions per soil layer, resulted in better results as expected. No significant differences were observed in $R^2$ values for features $EC_a$ uncalibrated and calibrated LIN or $rEC_a$. This may be due to the linear relationship between uncalibrated and

calibrated $EC_a$, and quasi-linear relationship with calibrated $rEC_a$, which does not add information to such variables





as polynomial features (Lavoué et al., 2010). However, the FDEM inverted $EC$ data generally underperformed the rest of features, with the only exception of the LS2 approach for $\theta$ prediction at Site 1, with a $R^2 = 0.19$ and a RMSE=0.047. While for Site 2 the maximum $R^2$ is 0.31, with a RMSE=0.066.

Comparing performances for predicting different soil properties, $EC_w$ was shown to be an easier target than any other
soil property. $EC_w$ prediction was generally better for Site 2 that does not have the influence of saline groundwater. Predicting $CEC$, clay content and $\rho_b$, on the other hand, seemed to be highly site-dependent.

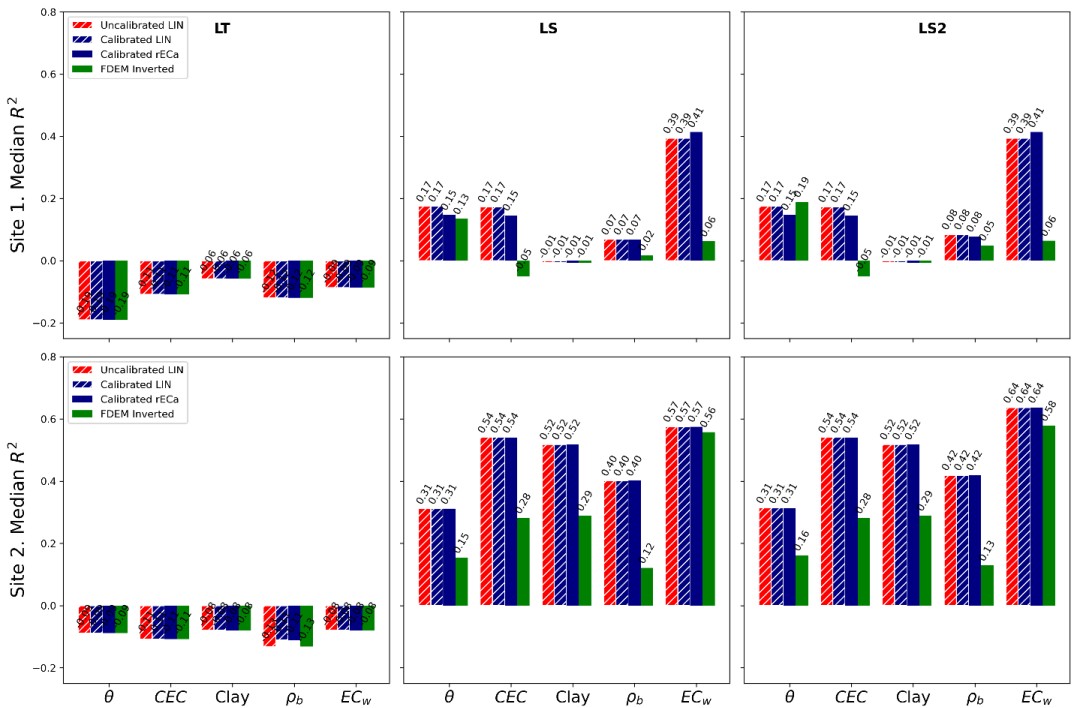

**Figure 6. Bar plots showing median results of stochastic modelling of soil properties. The median $R^2$ are obtained after testing such models in test datasets randomly generated as 30% of the original dataset and iterating 100 times to ensure**
**good data distribution.**

Subsequently, best performant stochastic models for $\theta$ prediction were implemented using the whole dataset (see Figure 7). Naturally, such model implementation shows a higher performance than the ones in Figure 6, because in Figure 7 the test is done on the same dataset used to develop the models However, evaluating a stochastic model on
the same dataset used for development is a improper practise that overestimates the performance of such model (Altdorff et al., 2017; Lipinski et al., 2008; Tibshirani et al., 2001). Then, implemented model errors should not be confused with the actual expected accuracy of a new $\theta$ sample prediction. Residuals of implemented stochastic models were not correlated with other soil properties.





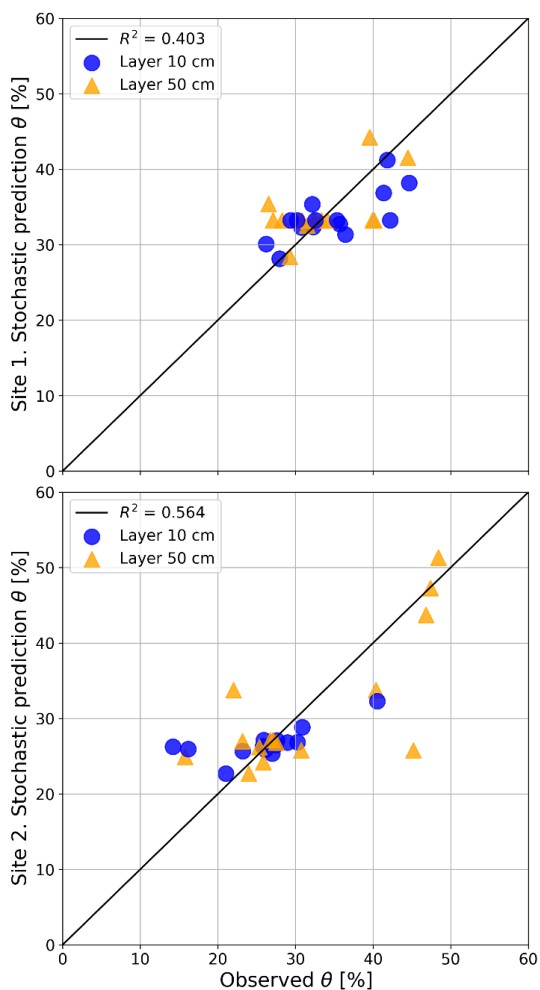

**Figure 7. Stochastic model implementation. Best performant stochastic models for $\theta$ prediction (based on Figure 6 results) were implemented using the 30 samples per site. Top subplot shows the data for Site 1, and bottom subplot shows data for Site 2.**

### 3.1 Sensitivity analysis

The result of the sensitivity analysis is presented in Figure 8 for Site 1 (upper subplot) and 2 (lower subplot). Generally, no possible combination of parameter values yielded an RMSE lower than 11% (or 0.11 cm3/cm3) for $\theta$ predictions, which corresponds to a negative $R^2$ value; that is, they performed worse than a single mean solution. Additionally, the predictions for $\theta$ were worse for Site 2 than for Site 1, presumably due to the larger variance in θ data at Site 2. The standard solution $\theta_0$ obtained using the $X_0$ parameter values was poorly performant too (see red lines in both subplots





of Figure 8). From Figure 8, the boxes which differ the most from the rest represent the most sensitive parameters. For both sites, the most sensitive parameters are the minimization method used and the pedophysical model approach.

Using the minimization method ROPE leads in general to better $\theta$ predictions, despite its average inversion error ($R^2$= 0.64 for Site 1 and $R^2$= 0.19 for Site 2) is higher than for Gauss-Newton ($R^2$= 0.75 for Site 1 and $R^2$= 0.94 for Site 2). Also, about 75% of ROPE inversions for both sites did not converge or reached a negative $R^2$ error, while for

Gauss-Newton most of inversions converged with a positive $R^2$.

The optimal approach in deterministic modelling is not the same at both sites. While the ID approach was the best in Site 1, the best in Site 2 was LT. This could be because ID uses actual soil properties to populate the pedophysical model (focusing on variance of the error), and the LT approach uses average soil properties (attacking the bias error), resulting in an unclear benefit because of the general poor performance.


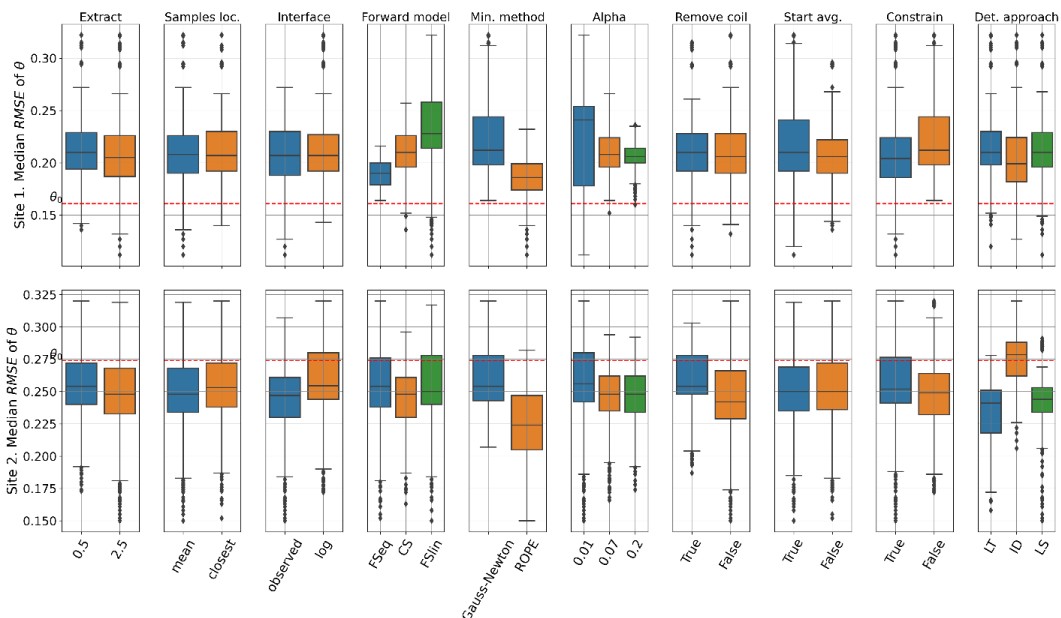

**Figure 8 . Box and whiskers plot results of uncertainty analysis. The figure shows the error outcomes of an elementary effects sensitivity analysis using parameters involved in processing all FDEM data. The top row displays results for Site 1, and the bottom row for Site 2. The plotted data is the median of RMSE. The error associated with the $\theta_0$ solution is**

**highlighted in red. Each box represents 50% of the data (i.e., the error associated with a specific parameter value) with a horizontal line indicating the median, while the whiskers represent 25% of the data at each end.**

## 4 Limitations

Although the presented research focuses on comparing different choices made along modelling steps, it is important to highlight its site-specific nature (Boaga, 2017). Therefore, because both sites were selected based on their



heterogeneous nature, the challenge that they represent is not necessarily representative of most common fields where the FDEM technique is applied, where collected $EC$ FDEM data normally have a narrower range (Minsley et al., 2012; van der Kruk et al., 2018).

While several modelling parameters were tested, the data acquisition strategy was not changed, and new findings can be obtained by, for instance, using a different algorithm to choose the sampling locations (Brus, 2019), or reducing the FDEM crossline sampling density to have closer matches between FDEM data and ERT and sampling locations. Also, the evaluation of different parameters in the sensitivity analysis was not exhaustive, with its results being relative to the parameters chosen.

For instance, using different optimization methods would improve the FDEM inversion error and offer more flexibility in the inversion, such as allowing variable layer thicknesses. However, not all optimization methods, such as the Gauss-Newton method, support variability in subsoil layer depths. Additionally, only 1D forward and inversion models using FDEM methods were employed, without considering lateral smoothing through 2D or 3D inversions. Furthermore, three different deterministic modelling approaches were tested using Equation 2, but other pedophysical

models were not considered. The difficulty in obtaining the $EC_s$ parameter of Equation 2 led to its exclusion, which might have compromised the model's effectiveness.

Lastly, the study was limited to univariable stochastic modeling. Multivariable regression incorporating more than one feature (such as using inverted $EC$ and uncalibrated $EC$), as well as other machine learning methods were not explored.


## 5 Conclusions and suggestions

Absolute soil property quantification using the FDEM method in heterogeneous fields is far from being accurate and methodologically solved.

The classical field-specific stochastic modelling of soil properties even limited, still offers the most straightforward solution. Based on our cases, uncalibrated $EC_a$ data can be used without compromising the effectiveness of such an approach. This bypasses the issues of physics-driven deterministic modelling, such as data calibration, robust $EC$ estimation, geophysical inversion, and pedophysical modelling. Such stochastic models should consider vertical soil variability, otherwise large mispredictions are expected. However, this is at the cost of building a dataset by sampling

and analyzing the soil target properties at the desired exploration depth. In samples not used for training, water content predictions achieved a poor best estimation with an R² value of 0.31 (RMSE=6.6%), which may be inadequate depending on the application, whereas $EC_w$ was better estimated.

In the case that sampling is not an option, a universal deterministic approach can be followed at the expense of FDEM calibration data, e.g. through ERT. A comprehensive sensitivity analysis of this approach shows that no possible

combination of modelling parameters could currently lead to reasonable predictions of water content for the studied sites. Particularly, the pedophysical model of Equation 2 should be reworked and validated for soil samples above 30% of clay content, and a pedotransfer function for $EC_s$ would help to ease its implementation. Additionally, the minimization method implemented in geophysical inversion turned out to be of key importance. Thus, and further work is required to improve the deterministic modelling predictions.






**Code availability**

https://github.com/orbit-ugent/FDEM_quantitative_soil

**Data availability**

https://zenodo.org/records/13465721

**Sample availability**

**Author contribution**

PS, MV and GB designed the surveys and carried them out. MV, GB and PS developed the model code and performed the simulations. EV contributed to statistical formal analysis. JV and WC gave writing advise. MV prepared the manuscript with contributions from all co-authors.

**Competing interests**

**The authors declare that they have no conflict of interest.**

**Acknowledgements**

Text readability and code performance were boosted by using generative AI (Chat-GPT 4o, OpenAI)

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
