# Peer review of "Quantitative soil characterization using frequency domain electromagnetic induction method in heterogeneous fields"

_EGUsphere, 2024_

## Author Response (AR2)

**RC1:**

The paper deals with the use of FDEM data for the quantitative characterisation of the first subsoil. The topic is of interest to the journal and the case study presented is a relevant example of an advance in the use of the EMI technique. Results are supported by the data presented and I encourage the publication. I suggest some minor revisions to be addressed before:

**Comments:**

**RC1.1**

- Ln 72 a ratio of the same quantity is by definition a-dimensional, so please remove ppm

**Response:** thanks for your suggestion. Despite indeed a ratio of the same quantity is a-dimensional, 'ppm' refers to parts per million ($1*10^{-6}$) and does not refer to a unit.

**RC1.2**

- some part of the paper cite ppm, other ppt (fig2), please homogenise it

**Response:** Right, this was corrected to ppm.

**RC1.3**

- All the equations labels have layout issue that do not agree with the journal editing rules

**Response:** all the equations were adapted to the journal's format

**RC1.4**

- Eq1 QP means the ratio of the imaginary part of primary and secondary fields ? Please specify

**Response:** this is now defined in the previous sentences. QP represents the imaginary component of the ratio between secondary and primary field.

**RC1.5**

- Ln 202-205 please provide references for the different polynomial approaches

**Response:** we reformulated this paragraph, now it reads:

'Three distinct approaches to polynomial development were utilized. A first approach, named "Layers Together" (ST-LT) consisted of combining data from different soil depths, so that no

differentiation was made between top- and subsoil samples for model development. Secondly, these sample sets were considered separately in an approach whereby different polynomials were developed for each soil layer ("Layers Separate"(ST-LS)). In this modelling approach, the same polynomial degree was maintained for both top – and subsoil data. Finally, the ST-LS2 approach was like ST-LS but permitted different polynomial degrees for the models of each layer.'

**RC1.6**

- Ln 232 please provide the used open-source software reference

**Response:** The reference was added. Mendoza Veirana, G., & Philippe De Smedt. (2024c). *orbit-ugent/Pedophysics: First release 0.1* (Version 0.1) [Computer software]. Zenodo. https://doi.org/10.5281/ZENODO.13465700

**RC1.7**

-Fig.5 To help reading I suggest to invert ideal and FDEM inverted EC columns, so reader can appreciate similarity between the former and the probe

**Response:** Fig 5 is updated

**RC1.8**

- Ln 304 missing dot before 'However' ?

**Response:** changed as suggested

**RC1.9**

- Ln 300-308 all this paragraph is not clear and should be re-written

**Response: changed as suggested:**

'When the best performing models for $\theta$ prediction are implemented using the entire dataset (**Error! Reference source not found.**) – using both training and test data – this outperforms the m odelling presented in **Error! Reference source not found.**, where only test data are incorporated in error assessment. While this is a common approach, we want to highlight this is improper practice to critically evaluate model performance as the inclusion of training data in error estimation results in an overestimation of model performance (Altdorff et al., 2017; Lipinski et al.,

2008; Tibshirani et al., 2001). In other words, implemented model errors should not be confused with actual expected accuracy of target property predictions.

To evaluate the influence of other soil properties in $\theta$ prediction, the residuals of the implemented empirical models were correlated with other soil properties, but these were not significant.'

**RC2:**

Dear authors,

I have read, your paper titled "**Quantitative soil characterization using frequency domain electromagnetic induction method in heterogeneous fields**" with interest.

The authors present a study, where frequency domain electromagnetic induction (FDEM), ERT and soil sampling measurements are used to evaluate the feasibility of FDEM modelling based on either a deterministic or stochastic approach to quantitative derive soil properties.

Overall, the paper is well structured and has a good readability. The topic is suitable for Hydrology and Earth System Sciences.

For enhances readability it would be helpful to add some tables or numeration as some points throughout the manuscript, see specific comments.

The methodology section needs refinement. There is an overwhelming amount of steps and I feel like there is some information on the test sites missing. Figure 2 is somewhat helpful but the steps shown in don't quite align with the text, e.g. the LIN transformation is missing from the text. Or use the same wording (maybe also in the section headers).

**Comments:**

**RC1.1**

- Line(s) 72: ppm – no abbreviation explanation (I think it is not necessary to explain but in Line 143, you explain ppt, so maybe be consistent)

**Response:** changed as suggested

**RC2.1**

- Line(s) 72: ppm – no abbreviation explanation (I think it is not necessary to explain but in Line 143, you explain ppt, so maybe be consistent)

**Response:** changed as suggested

**RC2.2**

- Line(s) 85 – 90: What are the land used for these test sites? Are the agriculturally used? Does a plow horizon exist? Was there any crop present while the measurements were performed? I feel

like in Figure 1 A, there can be two different field be disguised? The northwest part of the site seem the show a higher rECa compared to the part south of it.

**Response:** additional information about the sites was added.

**RC2.3**

- Line(s) 93: When you say driving speed. Was the device pulled with a tractor, quad? How did you ensure the distance from the ground?

**Response:** 'Field surveys at both sites involved collecting FDEM data using different sensors, all operating at 9 kHz: the Dualem-421S at Site 1 with a 3 m crossline sampling density, and the Dualem-21HS at Site 2 with a 1 m crossline sampling density, both with a constant distance above ground of 0.165 m using a sled. The instrument was pulled with a quad across the field with a driving speed of approximately 10 km/h, and a measurement sampling rate of 10 Hz.'

**RC2.4**

- Line(s) 94: By crossline density you mean crossline sampling density? Maybe use consistent wording here.

**Response:** changed as suggested

**RC2.5**

- Line(s) 96: Is PRP the same as VCP (vertical coplanar loops)? If yes I think VCP is more commonly used.

**Response:** it is not the same

**RC2.6**

- Line 96 – 99: May it is worth putting this into a table for a better overview?

**Response:** while I think this can add visibility, it would be too short as to be worthy.

**RC2.7**

- Line(s) 101 – 102: So there were multiple measurement? Could elaborate on that, when where they measurement and how often. The exact would be important to get a glimpse of the overall water content (summer cs. Winter)

**Response:** I think that this is not that relevant, because we are not aiming to compare seasonal variations, as the previously collected FDEM data was not complemented with soil sampling.

**RC2.8**

- Line(s) 121 – 122: Maybe use an enumeration? E.g., i.), ii.)?

**Response:** it is a very short listing, so we opted to keep it without enumeration. Also, across the text these steps are recalled by their name, which add meaning to the context, more than mentioning 'step iii)' for example.

**RC2.9**

- Line(s) 122: Does the pedophysical model have a certain name?

**Response:** yes, and it I mentioned later. Basically because the pedophysical modelling step it is not reduced to a single pedophysical model use.

**RC2.10**

- Line(s) 122: Line(s) 142: The combining of the ERT and FDEM data is done on which step in the flow chart in Figure 2?

**Response:** thanks for the observation, this was explicit just in the caption of the Figure 2, while now it is mentioned in the main text.

**RC2.11**

- Line(s) 144: LIN was introduced earlier already. Is LIN approximation the same as LIN transformation (Figure 2)

**Response:** yes. The double definition was corrected.

**RC2.12**

- Line(s) 146: How does shortening the profiles have an influence on the number of profiles?

**Response:** 'On the other hand, the inverted ERT $EC$ data was firstly grouped by profiles and any profiles at the beginning or end of each transect that did not reach a minimum depth of 4 m were removed due to the lower sensitivity in those peripheral zones.'

The readability was improved, 'shortening' was not suitable but 'removing'

**RC2.13**

- Line(s) 155: This figure already shows results but the section is located in the methodology section. May be move to results and discussion section?

**Response:** because this is a well stablished procedure and we do not expect insights from this figure, we kept it in the methodology. Also, this is not discussed further.

**RC2.14**

- Line(s) 199-201: But aren't soil probes multiple times represented? Maybe elaborate a little on this, for readers who are not too familiar with this method.

**Response:** soil probe readings are unique for each sampling point (n=30) and provide information along the soil analysis.

**RC2.15**

- Line(s)208: abbreviation CEC has not been mentioned before

**Response:** changed as suggested.

**RC2.16**

- Line 270: This is only a suggestion. Maybe it is easier to compare the different ways to derive the EC when they are in figure. You could split for the two soil layers and then use different Markers for the analysis types?

**Response:** thanks. As it is currently shown, the EC is highlighted more than the soil horizons. We think that this visualization put the focus on the discussion's goal. While if we mix the EC and separate by layer the visualization would be trickier.

**RC2.17**

- Line(s) 275 – 280: Is valid also throughout the manuscript. I would recommend using the wording of stochastic and deterministic modeling in connection with the different ECs. To remind the reader that this is the overarching goal to compare the two and to derive the feasibility.

**Response:** in this case, the naming is unmodified.

However, across the manuscript, 'stochastic' is changed by 'empirical'. Because the approach is based on data (empirical) but gives always the same result, since there are no stochastic processes (implying probabilities). This was realized thanks to discussion with colleagues.

**RC2.18**

- Line(s) 273: why not show this in the figure?

**Response:** because it is impossible to visualize that trend in this figure. A new one would be needed, but we do not find it that relevant. Although it is shown in the code.

**RC2.19**

- Line(s) 300. In the lower left plot the numbers overlapping with the bars are very hard to read. Move them above the bars, as the other subplots.

**Response:** changed as suggested.

**RC2.20**

- Line(s) 310: Why does it say in the legend 'Layer 10 cm' & 'Layer 50 cm'? Do you mean topsoil and subsoil.

**Response:** now the figure shows topsoil and subsoil legends

**RC3:**

The manuscript "Quantitative soil characterization using frequency domain electromagnetic induction method in heterogeneous fields" by Mendoza Veirana et al., under open discussion for a publication in HESS, deals with the difficult task of providing a quantitative use of frequency domain electromagnetic (FDEM) method. The author propose a novel and interesting framework to achieve this goal and show fairly convincing results that are, in my opinion, worthy of publication in HESS. My main comment of this work is the lack of discussions regarding the methods' resolution/footprint. The rest of the comments are rather minor and detailed below.

Damien                                                                                          Jougnot
CNRS                                    Senior                                              researcher
Sorbonne University

**Response:**

Thank you very much for your review and valuable insights. We acknowledge that the manuscript lacked a detailed discussion regarding the resolution and footprint of the FDEM method. In response, we have introduced this aspect as a limitation in the revised manuscript. After evaluating the effect of scale disparity in the empirical approach, we found that while it contributes to uncertainty, it is less critical than errors arising from the inversion process. This point has also been incorporated into the conclusions to clarify its relative significance. Thank you again for highlighting this important aspect.

**Comments:**

**RC3.1**

- Abstract: it is more usual to the abstract in a single (or two maximum) paragraph.

**Response:** changed as suggested

**RC3.2**

- Introduction: In my opinion, one limitation which not mentioned (or not explicitly enough) is related to the method footprints/resolutions. It is a tricky problem to solve, but when comparing sample characteristic to geophysical field-data, one always have to deal with the curse of scale (see, for example, the seminal paper of Day-Lewis et al., 2005). Every researcher looking for quantitatively interpreting field measurements is facing this issue of comparing things at different scales (from sample to the geophysical pixel, but also within a tomogram itself). I'm not pretending that I have a way to tackle that (I did try to study these mesoscopic heterogeneities at a very controlled scale with the highest possible resolution, I could only show that in heterogeneous media, petrophysical relationship are not always valid and should be considered with care, e.g., Jougnot et al. 2018), but I think that this kind of limitation/issues should also be mentioned in the introduction and later discussed in the paper.

**Response:** thanks for the detailed comment. Indeed, this is now mentioned in the introduction as a limitation of the method. Discussing the results of the empirical approach, we state:

'Comparing performances for predicting different soil properties, $EC_w$ was shown to be an easier target than any other soil property. $EC_w$ prediction was generally better for Site 2 that does not have the influence of saline groundwater. Predicting $CEC$, clay content and $\rho_b$, on the other hand, seemed to be highly site-dependent.

An important consideration when interpreting the empirical approach is the disparity in scale between the soil volumes measured by FDEM (cubic meters) and the smaller volumes analyzed in the laboratory (100 cubic centimeters). While some studies have addressed this issue, their conclusions are often site-specific and inconsistent (see e.g., Cong-Thi et al., 2024; Dimech et al., 2023). Despite these discrepancies, it is worth noting that non-inverted $EC$ generally outperforms inverted $EC$ for soil properties prediction, even though inverted $EC$ represents a smaller soil volume (cubic decimeters). This performance gap may be attributed to errors in the $EC$ inversion process rather than the difference in spatial scales.'

Which is now also mentioned in the conclusions.

**RC3.3**

- Line 46-47: I suggest additional reviews dedicated to soil electrical properties: Samouëlian et al. (2005) and/or Friedman (2005).

**Response:** changed as suggested

**RC3.4**

- Methodology: I really enjoyed the methodology chosen by the authors, I find it very robust and state-of-the-art.

**Response:** thanks a lot, pointing out the goods of a research also helps to progress of the author's skills.

**RC3.5**

- Line 80: It could be nice to provide examples/references of the fact that many soils violate the low induction number requirement. Showing that such framework is useful, especially when dealing with soils containing a clay fraction and trying to estimate this fraction.

**Response:** there are references for this. Also, because a robust transformation is also used, we skip showing soils as counter examples.

**RC3.6**

- Line 105: Why not measuring the electrical conductivity of the samples ? Since the authors analysed a lot of interesting and relevant properties, it would have been fun to test petrophysical relationship (like the one used later in the manuscript).

**Response:** the HydraProbe measurements aim to provide with this.

**RC3.7**

- Line 134-135: ResIPy is a great tool and the authors could provide a link in addition to the reference, especially for the Jupyter Notebook.

**Response:** we opt for mentioning the reference and not to provide a direct link, which indeed can be properly find.

**RC3.8**

- Figure 3: the authors should provide letters for the subplots.

**Response:** we did not provide letters for the subplots because these are not mentioned individually, the discussion here is short.

**RC3.9**

- Figure 4: the fact that rECa is systematically above the forward ERT model could be discussed, it is not clear to me why (possibly later in the paper)?

**Response:** this statement was added to the figure's discussion:

'. The $rEC_a$ consistently exceeds the LIN $EC_a$ reflecting the known limitations of the LIN approach and aligning with the tests reported by Hanssens et al. (2019).'

**RC3.10**

- Line 314: the subsection number should be 3.4

**Response:** changed as suggested

**RC3.11**

- Subsection 2.3.4: I am more of a deterministic guy but the stochastic approach described here is clearly interesting.

**Response:** thanks. I am more a well-empirical guy

**RC3.12**

- Section 4: As mentioned before, another strong limitation that should, in my opinion, be discussed, is related to the footprint differences between the methods (sampling, probe, ERT and FDEM). The water saturation or the clay content defined at the sample scale are likely to be different from the probe or geophysical method footprint scale (even without considering the smoothing related to inversion procedure). I think that this should be discussed as a limitation.

**Response** added to the above discussions, we state:

'Furthermore, the effects of scale disparities among soil sampling, ERT, FDEM, and probe measurements were not examined, even though they can introduce additional uncertainties when interpreting subsurface properties.'

**Refrences:**

Day-Lewis, F. D., Singha, K., & Binley, A. M. (2005). Applying petrophysical models to radar travel time and electrical resistivity tomograms: Resolution-dependent limitations. *Journal of Geophysical Research: Solid Earth*, *110*(B8).

Friedman, S. P. (2005). Soil properties influencing apparent electrical conductivity: a review. *Computers and electronics in agriculture*, *46*(1-3), 45-70.

Jougnot, D., Jiménez-Martínez, J., Legendre, R., Le Borgne, T., Méheust, Y., & Linde, N. (2018). Impact of small-scale saline tracer heterogeneity on electrical resistivity monitoring in fully and partially saturated porous media: Insights from geoelectrical milli-fluidic experiments. *Advances in Water Resources*, *113*, 295-309.

Samouëlian, A., Cousin, I., Tabbagh, A., Bruand, A., & Richard, G. (2005). Electrical resistivity survey in soil science: a review. *Soil and Tillage research*, *83*(2), 173-193.